# Demonstration of Tunable Shielding Effectiveness in GHz and THz Bands for Flexible Graphene/Ion Gel/Graphene Film

Lixiang Yao , Yuanlong Liang, Kui Wen, Yanlin Xu *, Jibin Liu, Peiguo Liu and Xianjun Huang *

College of Electronic Science and Technology, National University of Defense Technology, Changsha 410005, China; j94723082@163.com (L.Y.); liangyuanlong18@nudt.edu.cn (Y.L.); kuiwen93@hotmail.com (K.W.); liujibin@nudt.edu.cn (J.L.); pg731@126.com (P.L.)
* Correspondence: 13298656824@163.com (Y.X.); huangxianjun@nudt.edu.cn (X.H.)

**Abstract:** To satisfy the demands of wireless communication systems for tunable shielding materials, in this work, a graphene/ion gel/graphene sandwich structure is proposed, based on graphene and a solid ionic material ion gel. After modelling, preparing and testing, it was found that the structure could achieve more than 10 dB shielding effectiveness tuning in GHz and THz bands. Meanwhile, the adjusting speed of the structure was also studied, displaying effective dynamic tuning in the second order. Furthermore, the fabricated samples have the advantages of a low profile, easily conformable, and convenient processing, which are of great potential in emerging electronic devices.

**Keywords:** shielding effectiveness; tunable; graphene; ion gel; multiband

## 1. Introduction

Rapidly developing wireless communication systems have brought the urgency of protecting devices from electromagnetic interference (EMI) [1–3]. Generally, shielding materials, metal and composite materials, for example, are widely used to isolate unexpected signals and avoid radiating interference signals to free space, which is essential for improving the electromagnetic compatibility (EMC) of devices [1,4–6]. However, the fixed physical dimensions and electrical properties of those materials also limit their ability in adapting to increasingly intelligent electronic systems [7]. Therefore, it is imperative to develop the shielding effectiveness (SE) of tunable materials, which could realize the precise control of electromagnetic distribution in time and space [7–9]. In the GHz band, where the shielding of communication signals is primary, by switching the SE of shielding materials between blocking and transmitting, the space utilization of places like confidential meeting rooms, airport security chambers or military duty rooms can be greatly improved. In the THz band, where effective generation, detection and modulation methods of THz signals are still scarce, the use of low profile, structurally stable and tunable EMI shielding materials can effectively realize the wavefront modulation of terahertz electromagnetic waves [10–13]. Besides, the flexibility and conformability of the materials are also key factors in practical applications, which are of great significance for foldable displays, micro robots, and wearable devices [2,14].

At present, the implementation of tunable SE mainly includes the loading of active devices between periodic arrays, the use of electromagnetic characteristics, tunable materials, or the change of structural parameters, and there are few reports on materials having tunable characteristics both in the THz and GHz frequency bands [7,9,15–17]. By loading active devices such as MEMS or PINs, electromagnetic waves can be shielded in a specific frequency range [17–19]. However, the frequency selectivity of this method is too strong to achieve broadband control, and the array-type point loading method brings more complex near-field effects, limiting the miniaturization and compact application of devices. The most widely used method to achieve tunable SE is to add conductive fillers, such as carbon nanoparticles, to the polymer to obtain a polymer composite shielding material,

by changing the aspect ratio, conductivity, permeability or other parameters of the filler through external conditions, the SE could be adjusted [7–9,16,20].

In recent years, with further research on graphene, a two-dimensional material with adjustable electrical properties, a variety of graphene photoelectric modulation units have been born, which can realize the regulation of SE in a wide frequency range [2,21,22], especially in the terahertz frequency range [11,13,22–24]. The authors of [25] proposed a broadband SiO$_2$-based graphene terahertz modulator, by applying voltage from 0–50 V, it realized the tuning of transmission amplitude for about 15% in the 0.57–0.63 THz band. Based on the supercapacitor structure, a graphene/ionic liquid/graphene sandwich structure was subsequently proposed to realize the photoelectric modulation [26–32], which could achieve a modulation depth of 50% between 0.1 and 0.4 THz with voltage less than 3 V [27]. However, the ionic liquid also limits the application scenarios of the structure to a certain extent, and has the disadvantages of poor weather resistance and difficulty in conformability, thus finding it hard to meet the urgent needs of flexible electronics.

In this paper, we propose the use of solid ionic materials to construct a graphene/ion gel/graphene (GIG) sandwich structure, which has excellent electronic and structural stability. Through modeling, processing and testing, the obtained samples with monolayer and bilayer graphene both could gain more than 10 dB amplitude tuning of SE in GHz and THz bands. Applying rectangular and triangle waves on the electrodes, the transmission coefficients of two samples in 7 GHz were also recorded in the time domain, displaying the corresponding waveforms as the bias voltages, whose time constants were calculated subsequently, showing effective dynamic tuning ability in the second order. Briefly speaking, the proposed GIG structure revealed great potential in the field of flexible electronics.

## 2. Principle Analysis and Modelling

### 2.1. Principle Analysis

According to the Kubo formula, the conductivity of graphene includes intra- and interband contributions, which can be described as [21,33–36]:

$$\sigma(\omega, \mu_c, \Gamma, T) = \frac{je^2(\omega - j\Gamma)}{\pi\hbar^2} \times \left[ \begin{array}{c} \frac{1}{(\omega - j\Gamma)^2} \int_0^\infty \varepsilon \left( \frac{\partial f_d(\varepsilon)}{\partial \varepsilon} - \frac{\partial f_d(-\varepsilon)}{\partial \varepsilon} \right) d\varepsilon \\ - \int_0^\infty \varepsilon \left( \frac{f_d(-\varepsilon) - f_d(\varepsilon)}{(\omega - j\Gamma)^2 - 4(\varepsilon/\hbar)^2} \right) d\varepsilon \end{array} \right] \tag{1}$$

where $f_d(\varepsilon) = [\exp(\varepsilon - \mu_c)/(k_B T) + 1]^{-1}$ is the Fermi−Dirac distribution, $\Gamma$ is the carrier scattering rate (its reciprocal $\tau = 0.2$ ps is the electron relaxation time), $\hbar$ is the reduced Planck constant, $T$ is the Kelvin temperature, $\mu_c$ represents the chemical potential of graphene, $\varepsilon$ is the energy, and $k_B$ is the Boltzmann constant.

By applying voltage, pumping light or ion doping, the chemical potential of graphene can be changed to achieve conductivity regulation, and the depicted conductivity curves of graphene changing associated with frequency and $\mu_c$ are as follows [33–35]:

It can be observed from Figure 1 that in the microwave frequency band and the low terahertz frequency band, the conductivity of graphene was independent of frequency and only changed with the chemical potential, thus the tuning of chemical potential can effectively control the conductivity of graphene.

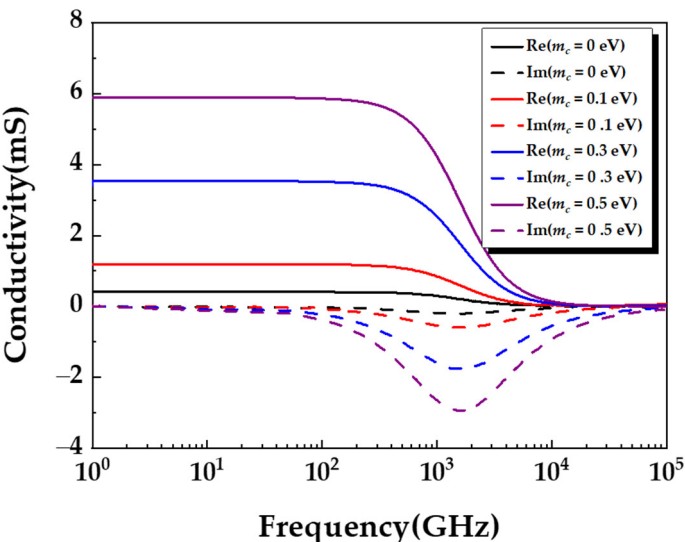

**Figure 1.** The conductivity of graphene with various chemical potentials and frequency.

## 2.2. Modelling and Fabrication

In this paper, a supercapacitor-based graphene/ion gel/graphene sandwich structure is adopted to achieve the tuning of graphene. Among these, the ion gel acts as a dielectric layer and the monolayer or bilayer graphene as electrodes, as shown in Figure 2a [37]. When voltage is applied to the edge of the electrodes, the Fermi level of the graphene changes accordingly, thereby realizing the regulation of its electrical properties.

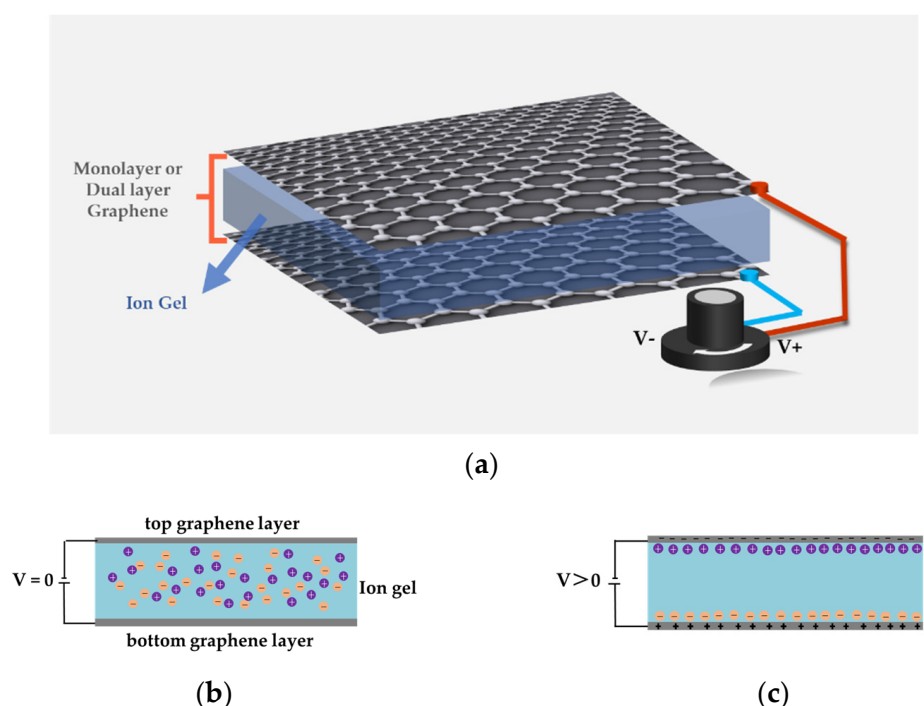

**Figure 2.** The diagram of GIG sandwich structure and regulation principle: (**a**) the schematic of GIG sandwich structure; (**b**,**c**) the distribution of charge, (**b**) without and (**c**) with bias voltage.

Ion gel is a solid mixture composed of polymers and salt electrolytes. Because of its abundant anions and cations, it has good thermal stability and excellent electrical properties, appearing gelatinous at room temperature [10,38,39]. With the action of an electric field, the distribution of cations and anions is uneven in Figure 2c, forming an

obvious electric double layer (EDL) at the interface. It is worth noting that the distance between the EDLs on the interface is much smaller than that of traditional capacitors, which will produce a larger capacitance. A unit area capacitance value in the order of uF/cm can be obtained at the thickness of the um level, so it can realize low voltage regulation of the graphene energy level [15].

In this paper, monolayer or bilayer graphene grown on 0.1 mm thick PET film by the CVD method (as shown in Figure 3a) was used to obtain the monolayer graphene/ion gel/monolayer graphene (MIM) and bilayer graphene/ion gel/bilayer graphene (BIB) samples, respectively, and conductive silver paste was printed on the edge of the graphene, as the electrode, using electrofluid printing technology, to reduce contact resistance, whose surface was then covered with copper foil to facilitate the tuning process. The ion gel was obtained by dissolving polyvinylidene fluoride-co-hexafluoropropylene (PVDF-co-HFP) and 1-ethyl-3-methylimidazole bistrifluoromethanesulfonimide ([EMIM][TFSI]) in a mass ratio of 1:4 in an acetone solution with a mass ratio of 7, and magnetically stirring at 60 °C until it became transparent [37,39]. Afterwards, the acquired ion gel was spin-coated onto the graphene film, then covered with another graphene film according to the model shown in Figure 2a to obtain the GIG structure. Finally, the sample was squeezed and coated using a rolling machine and placed in an oven at 50 °C for two hours. Figure 3b shows the obtained GIG sample, with the dimensions of 8 × 8 cm and total thickness of 0.3 mm, where the ion gel layer was about 50 μm thick, displaying the advantages of low profile and easy conformability. Furthermore, the sample also exhibited excellent structural stability and could be bent at a large angle.

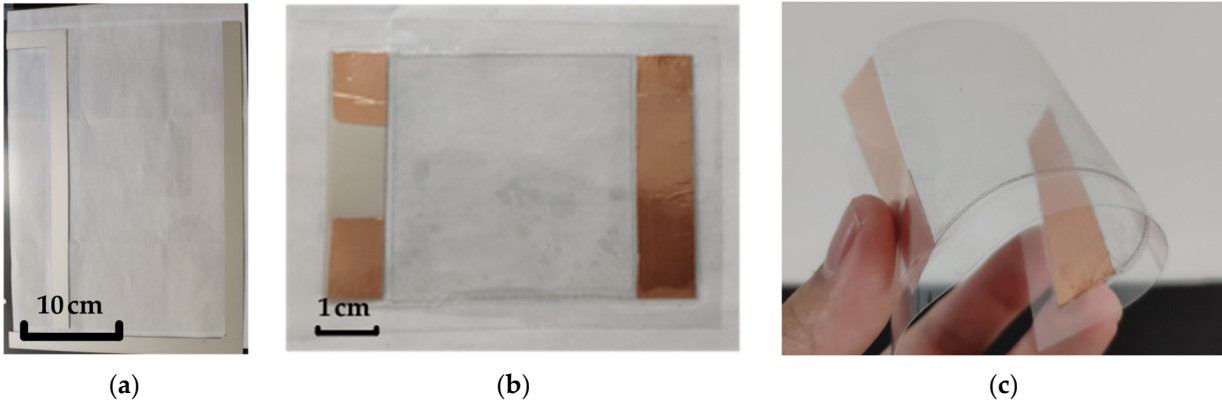

**Figure 3.** (**a**) The adopted CVD graphene grown on PET film; (**b**) the fabricated GIG sample (8 × 8 cm); (**c**) GIG sample bent at a certain angle.

## 3. Demonstration of Tunable Shielding Effectiveness in GHz Bands

### 3.1. Design and Experiments of Different Shielding Film Structures

3.1.1. Definition of Shielding Effectiveness

The shielding effectiveness of material is defined as the ratio of electric field strength, magnetic field strength or power in the free space before and after the shielding body is placed, which can be expressed as [40,41]:

$$\text{SE} = 20\lg(E_i/E_t) = 20\lg(H_i/H_t) = 10\lg(P_i/P_t) \tag{2}$$

According to the definition of the S parameter in a dual-port network, $S_{21} = (b_2/a_1)_{a_2=0}$, $S_{12} = (b_1/a_2)_{a_1=0}$, where $a_1, b_1, a_2, b_2$ represent the field strength of incident and reflected waves of port 1 and port 2, respectively. Thus, the SE of a material can be described by the transmission coefficient (T) as [41]:

$$\text{SE} = -20\lg|S_{21}| = -20\lg|S_{12}| = -10\lg(T) \tag{3}$$

The SE of graphene greatly depends on the position of its Fermi level. With the increase of bias voltage, doping between graphene levels leads to the increase of conductivity and the shielding effectiveness towards the electromagnetic waves.

### 3.1.2. The Waveguide Testing Method

In the microwave frequency band, it is difficult to directly test the S parameters of samples in a wide frequency range using the space−field method, so the waveguide method is widely used [23]. Besides, as learned in Figure 1, the conductivity of graphene in the microwave frequency band is independent of frequency. Therefore, based on the existing experimental conditions and sample size, we chose the frequency bands from 6 to 8 GHz as a representative case, which is widely used for the communication of satellites and television broadcasting and only a pair of waveguides are needed to obtain the SE properties of the sample.

The schematic and setup of the waveguide method are shown in Figure 4a,b, where the sample is placed between the waveguides, and the bias voltage is applied by the DC power supply connected to two electrodes. The transmission curves with different voltages are recorded on the vector network analyzer, and the SE curves can be acquired from Equation (3). Here we chose the standard waveguide WR-137 for the measurement, whose working frequency band is 5.38–8.17 GHz and the size of its test window is 34.849 mm $\times$ 15.799 mm. During the test, the operating mode of the electromagnetic wave in the waveguide was $TE_{10}$ mode, and the SE analysis was in the near field.

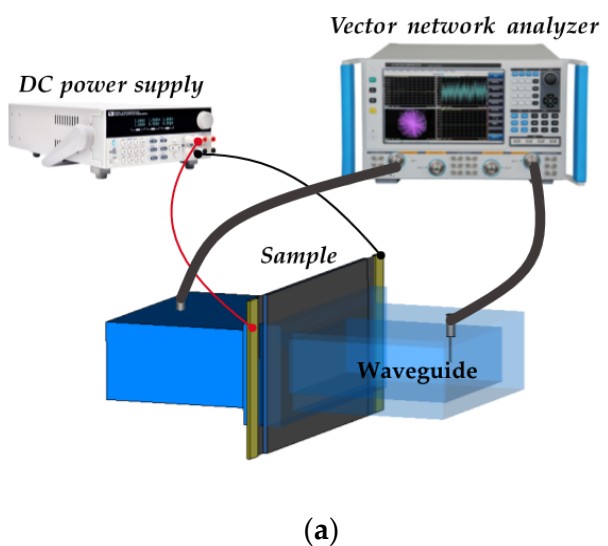
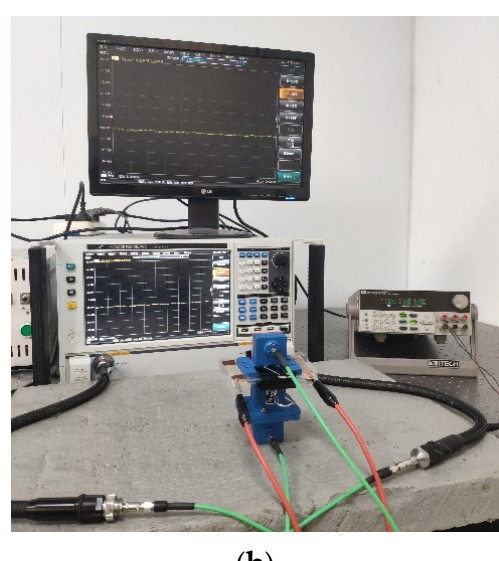

(**a**)  (**b**)

**Figure 4.** (**a**) The schematic and (**b**) experimental setup of waveguide testing method.

### 3.1.3. Experiments on Shielding Effectiveness of MIM and BIB Structure

For graphene grown by the CVD method, it is easy to be doped by various ions or impurities, when it is exposed to air. Without bias, the Fermi level is in the p-doped state and will not be located in the center of the energy band [10,26]. When the voltage is applied, the top and bottom graphene are doped with electrons and holes, respectively, as shown in Figure 5a. Therefore, as the voltage increases from the initial level, the shielding effectiveness will show a nonlinear and nonmonotonic change, as depicted in Figure 5b.

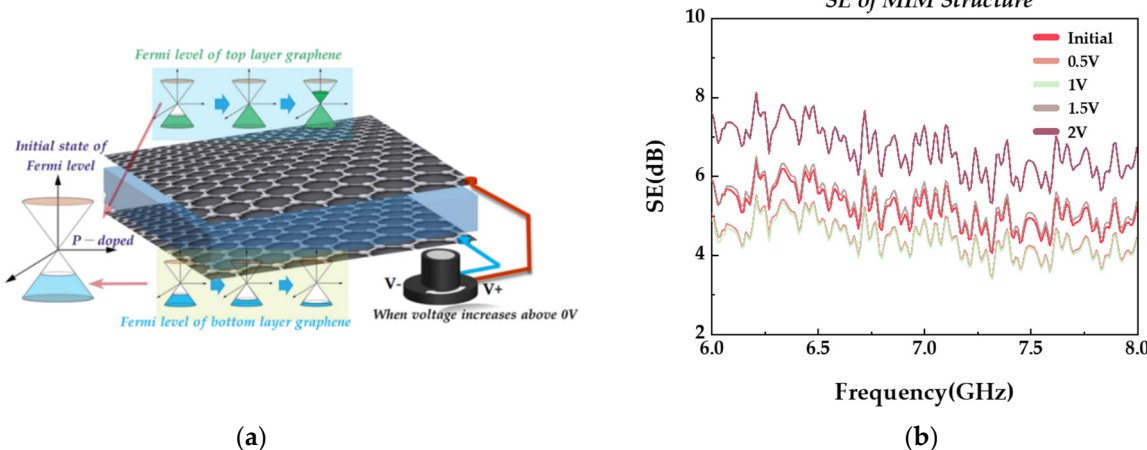

**Figure 5.** (**a**) The Fermi level of graphene under bias voltage; (**b**) SE curves of MIM structure when voltage increases from initial.

The applying of bias voltage will change the p-doped energy level state of the graphene. With the dynamic changing of the voltage from 0 to 6 V, the Fermi level of graphene achieved a dynamic balance from the low doped state in the band gap center to the high doped state. The SE of the samples accordingly showed a monotonous trend with the change of bias voltage. The tested SE curves of MIM and BIB structure are shown in Figure 6a,b.

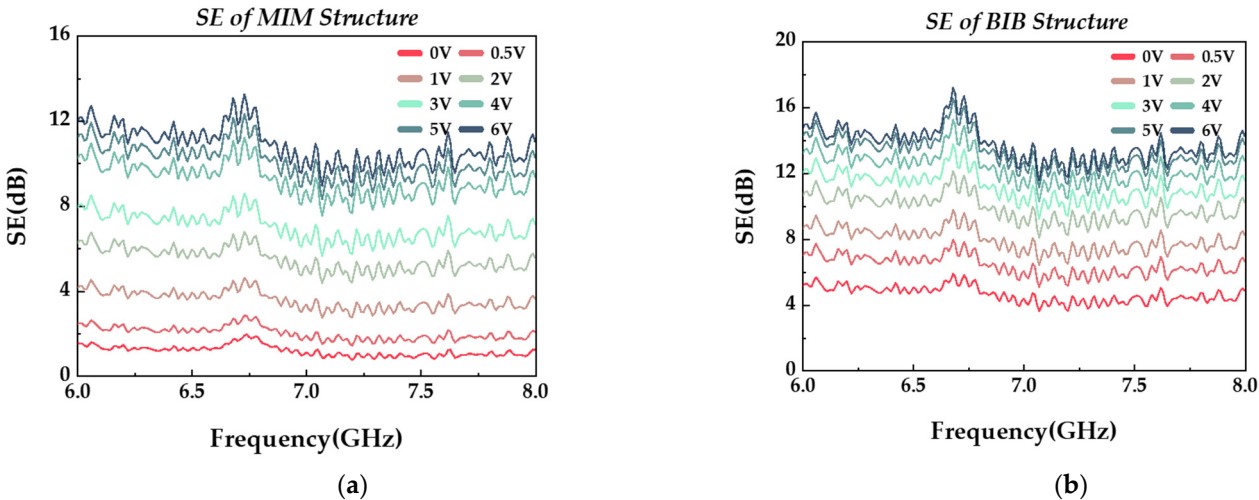

**Figure 6.** Shielding effectiveness of the (**a**) MIM and (**b**) BIB structure with different voltages.

The results show that both of the GIG samples could realize the tuning of shielding effectiveness for above 10 dB in the 6–8 GHz band, between which the MIM structure exhibited a slightly larger tuning range. For the BIB structure, with the voltage changing from 0 to 6 V, its SE was tuned from less than 5 dB to close to 15 dB, showing a stronger shielding effectiveness than the MIM sample.

### 3.1.4. Experiments on Shielding Effectiveness of Multilayer MIM Structure

As shown in Figure 7, four MIM samples fabricated by the same process were separated by PET films with a thickness of 0.35 mm, and the electrodes were connected in parallel, forming a cascade structure with a total thickness of about 2.25 mm. The results of the SE were tested and the curves were drawn as follows:

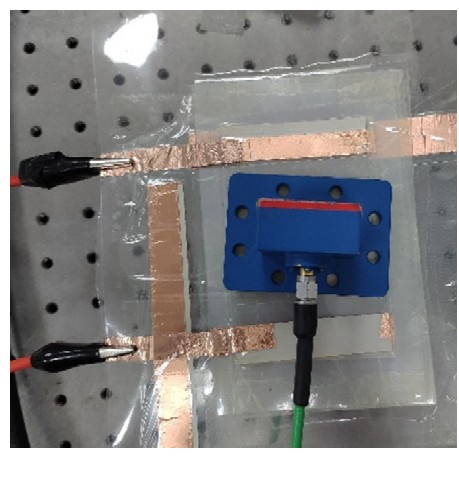 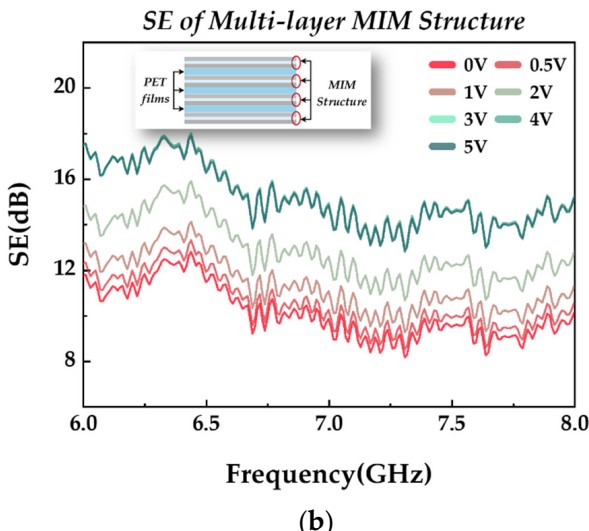

(**a**)            (**b**)

**Figure 7.** (**a**) The experimental photo of multilayer MIM structure; (**b**) shielding effectiveness of multilayer MIM structure with different voltages.

On the one hand, the minimum SE value of the cascade structure was much greater than that of a single GIG structure. On the other hand, the SE regulation range of the cascade structure was only 5 dB, which was significantly lower than that of the single structure. The main reason was that the uniformity of the ion gel in the four structures was different, resulting in different doping levels of the monolayer graphene under the same bias voltage.

### 3.2. Dynamic Control of Shielding Effectiveness

#### 3.2.1. Measurement Method

According to the previous part of this paper, applying voltage to both sides of the GIG structure will achieve the tuning of the Fermi level of graphene, and the transmission performance of electromagnetic wave changes accordingly. Therefore, when the bias voltage is a modulated signal with a certain waveform, the SE of GIG structure will also exhibit certain modulation characteristics in the time domain and realize the dynamic control of SE, as shown in Figure 8a.

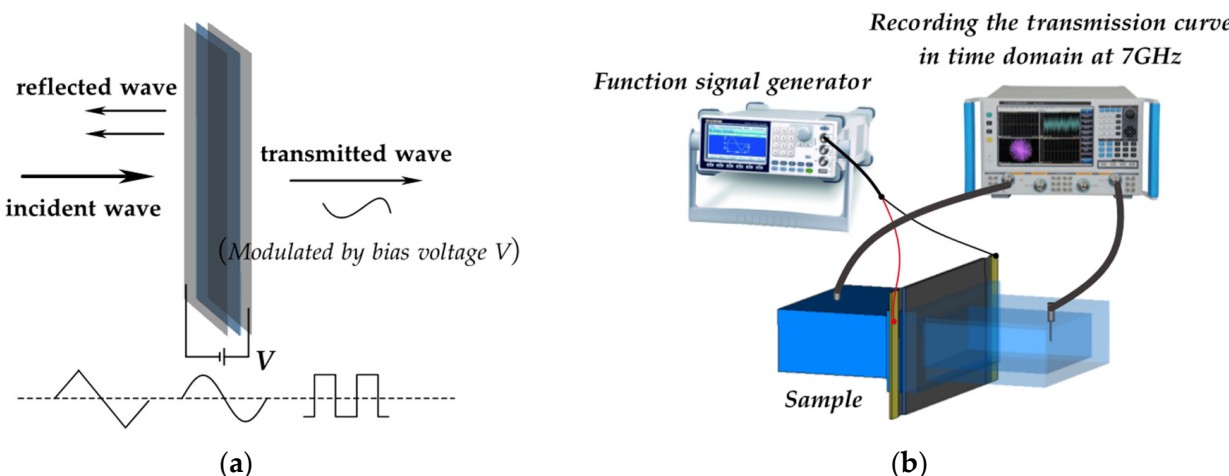

(**a**)            (**b**)

**Figure 8.** (**a**) The diagram and (**b**) the setup of the dynamic control on SE.

Figure 8b shows the setup of the modulation performance testing process of the structure by using the waveguide method. The sample is placed between the waveguides, and arbitrary signal generated by the function signal generator is applied to the electrodes. Meanwhile, the vector network analyzer is set to the point frequency scanning mode to record the transmission coefficient at 7 GHz in the time domain, which will exhibit certain similarities with the bias signal waveform.

Applying rectangular and triangle waves to electrodes, the transmission coefficient curves in the time domain of the MIM and BIB structure were then recorded respectively, as shown in Figure 9. Considering the symmetry of the structure, only the voltage above zero was tested.

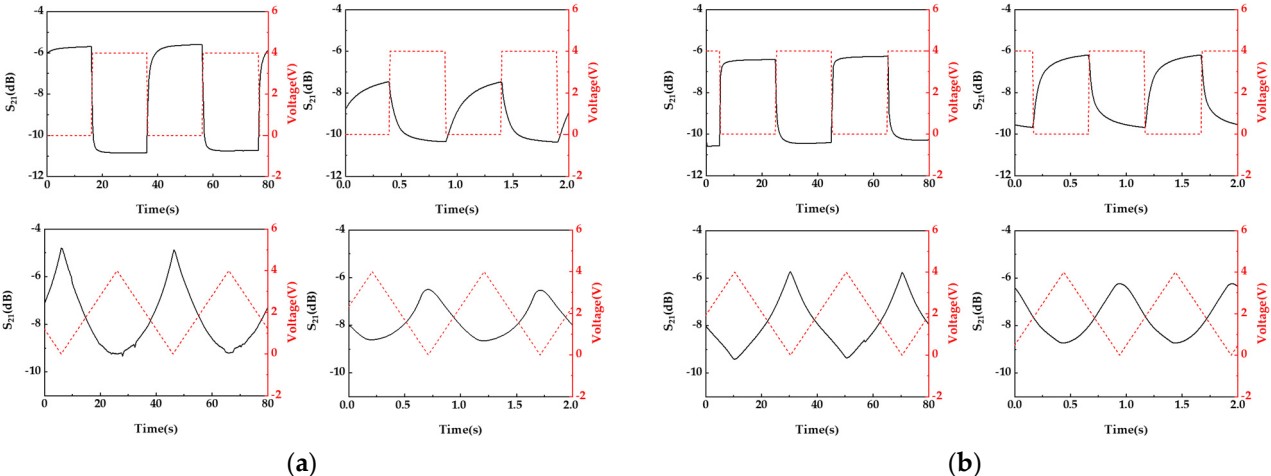

**Figure 9.** Transmission curves of (**a**) the MIM and (**b**) the BIB structure when applying rectangular and triangle waves with the frequency of 250 mHz and 1 Hz.

It can be observed that when the period of bias signal was in the second order, the sample could effectively tune the transmitted electromagnetic wave (amplitude change $\geq$ 3 dB). When the voltage amplitude increased, the transmission coefficient decreased, indicating that the shielding efficiency of the sample increased, and vice versa. With the increase of voltage frequency, the regulation range decreased gradually, and the sample with monolayer graphene exhibited greater attenuation.

Besides, when the voltage was switched between high and low levels, the transmission performance of the samples did not change transiently, contrarily, it changed relatively slowly, similar to the charging and discharging process of the capacitor. The main reason is that the mass and volume of anions and cations in an ion gel are much larger than those of the carriers in the graphene, and they move more slowly under the action of the electric field, which limits the regulation speed of the samples. Consistent to the case of rectangular wave bias, when the triangular wave was applied, the transmission curves of the two samples in the time domain showed a similar shape to the bias signal, where the nonlinear also mainly came from the slower moving speed of ions.

The experimental results show the different regulation ranges and speeds of the two samples with different layers of graphene, mainly due to their different carrier concentrations.

### 3.2.2. Response Time of Tunability

The modulation speed and performance of SE tunable materials mainly depend on the carrier concentrations in the dielectric layer. For the established GIG sandwich structure, its tuning speed was greatly limited by the ions in the ion gel [10,15,22]. As learned from the above results, the GIG structure could realize the effective tuning of the electromagnetic wave at second level.

Commonly, the time constant $\tau$ is often used to express the time process of a transition reaction for a certain quantity decaying exponentially. Thus, the time constant is introduced to quantitatively describe the ability of the sample to dynamically control the electromagnetic wave.

In this work, the transmission coefficient changing process of the GIG structure in the time domain can be divided into two stages: falling and rising, with the voltage switched between high and low levels, corresponding to the time constants $\tau_{fall}$ and $\tau_{rise}$, as shown in Figure 10, where $\tau_{fall}$ refers to the time required in the process of the transmission coefficient decaying from maximum to minimum for $(1-1/e)$ of the regulation range, and $\tau_{rise}$ represents the process of the transmission coefficient rising from minimum to the $(1-1/e)$ of the regulation range [10].

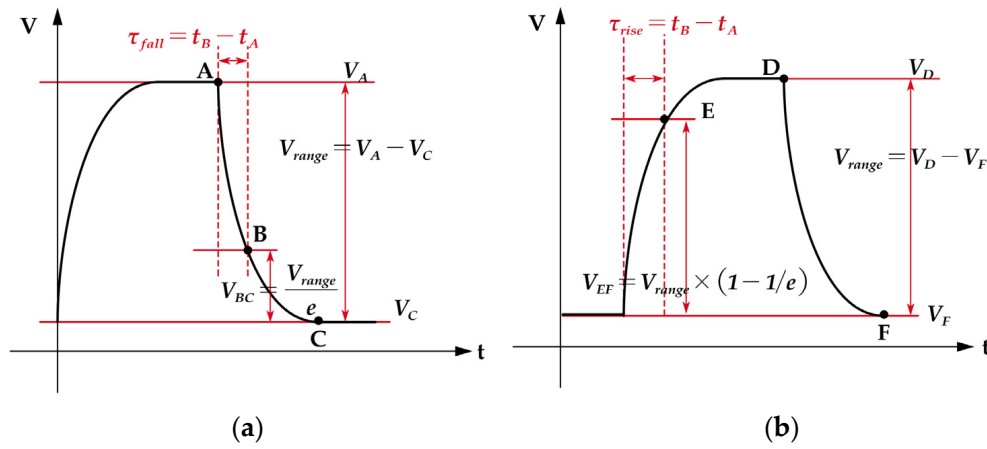

(**a**)            (**b**)

**Figure 10.** Definition of time constant in (**a**) the falling and (**b**) rising process.

Combined with the test results from Figure 9, the time constants $\tau_{fall}$ and $\tau_{rise}$ of the MIM and BIB samples were obtained, respectively, as shown in Table 1, where the MIM structure exhibited a faster speed in the falling process. However, when the voltage was switched to 0, the BIB structure spent less time to restore to the initial level. The main reason is that in the falling process, the charging time of the structure mainly depended on ions in ion gel, thus making the BIB structure need more time to dope the Fermi level of the bilayer graphene. When the voltage was switched to 0, the carriers in the graphene dominated, resulting in a smaller $\tau_{rise}$ for the BIB structure, which had a richer content of carriers.

**Table 1.** Time constant of two samples.

|  | $\tau_{fall}$ | $\tau_{rise}$ |
| --- | --- | --- |
| MIM structure | 0.578 s | 1.006 s |
| BIB structure | 0.654 s | 0.859 s |

## 4. Demonstration of Tunable Shielding Effectiveness in THz Bands

In the THz band, the conductivity of graphene is still greatly affected by its chemical potential as shown in Figure 1, thus making it possible to realize the tuning of shielding effectiveness. Meanwhile, it is worth mentioning that the generation, modulation and detection methods of the THz-electromagnetic wave are different from those in the microwave band, where the terahertz spectroscopy technique is needed [10,13,22,42], whose schematic and the test scenario of samples are as follows:

As shown in Figure 11a, the THz time-domain spectroscopy (THz-TDS) system is a THz generation and detection system based on coherent detection technology [42]. After obtaining the amplitude and phase information of THz pulse, it can directly obtain the absorption coefficient, refractive index and transmittance of the sample by Fourier

transform of time waveform. In this paper, the TPS spectra 1000 was used to test the transmission coefficient of the samples. As shown in Figure 11b, the sample is placed between two parabolic mirrors, and the calculated SE curves of the two samples under different voltages are depicted in Figure 12.

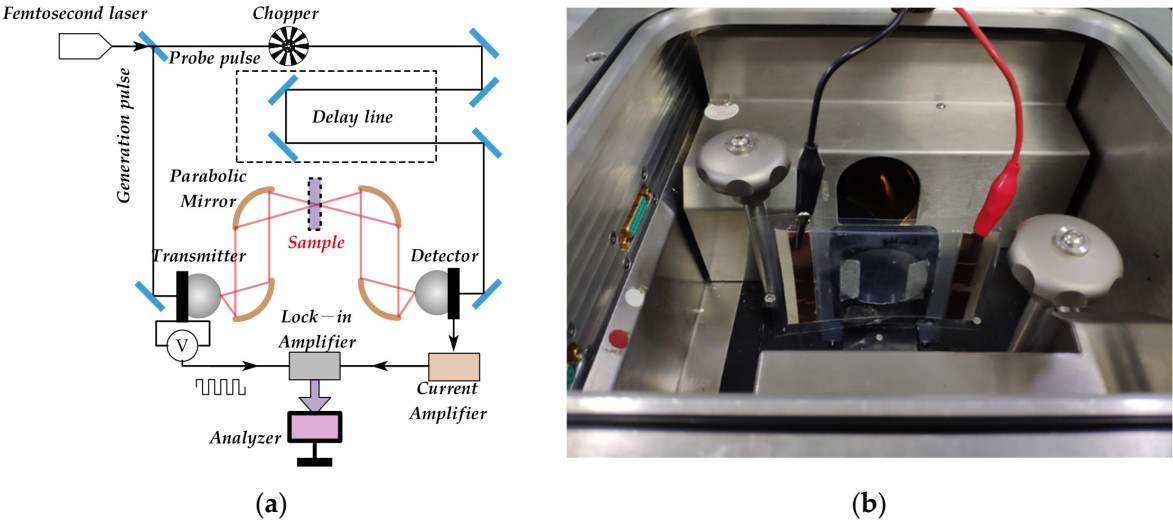

**Figure 11.** (**a**) The diagram of THz-TDS system; (**b**) sample placed between parabolic mirrors.

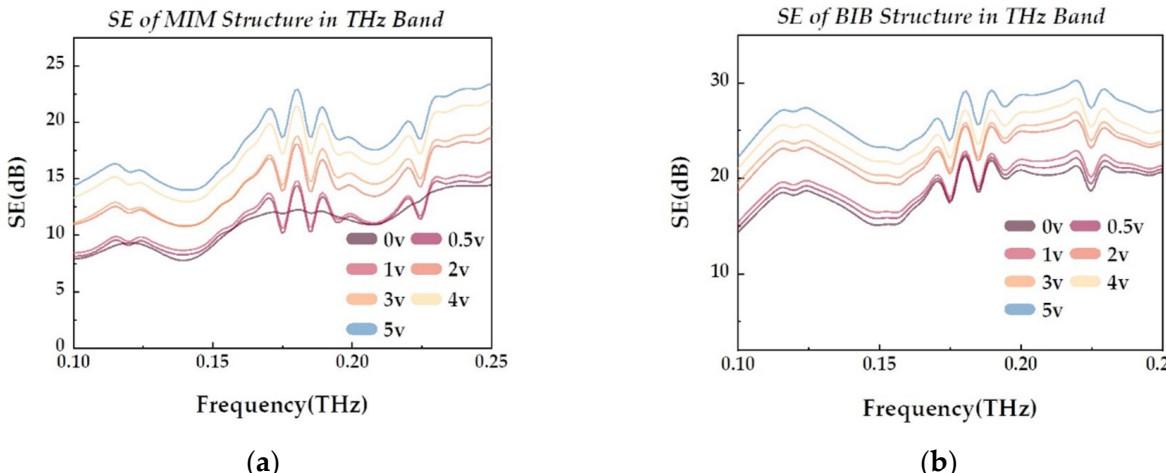

**Figure 12.** Shielding effectiveness of (**a**) the MIM and (**b**) BIB structures under different voltages in THz band.

The results show that with the voltage increasing from 0 to 5 V, the SE regulation range of the GIG structure could still achieve close to 10 dB in the low THz frequency band. For the BIB structure, it revealed a much higher shielding effectiveness than that of the MIM structure. In addition, it can also be observed that as the frequency increased, the SE values of the two samples increased consistently. The main reason is that for the uniform graphene film-type shielding material, it had a better shielding ability for high-frequency electromagnetic waves, similar to the skin effect of metal materials, thus having a higher SE value in the high frequency band.

## 5. Conclusions

In brief, a graphene/ion gel/graphene sandwich structure was modeled, fabricated and tested. In our work, CVD graphene transferred onto PET films were adopted and the ion gel was prepared in the laboratory by consulting the related references. During the test,

the waveguide method and the terahertz time−domain spectroscopy system were used, respectively, to obtain the shielding effectiveness of the samples in GHz and THz frequency bands. The results showed that the GIG structure could achieve at least a 10 dB effective low-voltage tuning of SE in 6–8 GHz and 0.1–0.25 THz bands under the different conditions of monolayer and bilayer graphene, presenting stable broadband tuning performance. Besides, the regulation speed of samples in 7 GHz was also studied by applying rectangular and triangle waves with certain frequency on electrodes, displaying an effective dynamic control of the transmitted waves in the second order. Meanwhile, in addition to excellent electrical performance, the fabricated samples also have the advantages of low profile, easy conformability, superior structural stability and flexibility with a convenient fabricating process, leading to great prospects in the field of flexible electronics.

**Author Contributions:** Conceptualization, L.Y., Y.X., J.L., P.L. and X.H.; methodology, L.Y., Y.L. and X.H.; validation, L.Y. and Y.L.; writing—original draft preparation, L.Y.; writing—review and editing, L.Y., K.W. and X.H.; funding acquisition, Y.X., J.L., P.L. and X.H. All authors have read and agreed to the published version of the manuscript.

**Funding:** This work was supported by the National Natural Science Foundation of China (No.61801490) and Research Project Foundation of National University of Defense Technology (No.ZK18-03-15).

**Institutional Review Board Statement:** Not applicable.

**Informed Consent Statement:** Not applicable.

**Data Availability Statement:** Not applicable.

**Conflicts of Interest:** The authors declare no conflict of interest.

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
