# Peer review of "Demonstration of Tunable Shielding Effectiveness in GHz and THz Bands for Flexible Graphene/Ion Gel/Graphene Film"

_applsci, doi:10.3390/app11115133_

Round 1

Reviewer 1 Report

The article does not have the typical IMRAD structure of a scientific article and can therefore be difficult to read. It seems to me that maintaining such a structure is important due to a certain standardization for which potential recipients of the article are prepared. Particularly important is the lack of an extended Results and Discussion chapter. Currently, the article is in the form of a research report from two tests carried out in two frequency bands - GHz and THz. It also seems unfortunate to me that two chapters (3 and 4) are entitled Demonstration of .... The aim of a scientific article is not to demonstrate something but to present well-established, simulation, experimentally and analytically verified results of research and experiments. The statement "Demonstration" only reassures the reader that this is a report from part of an ongoing study.

Detailed comments:

  1. Fig. 1 is too small and therefore not readable, especially when it comes to the elements of the legend.
  2. How does the shielding properties change as a function of the thickness of the Ion gel layer? Especially for external AC voltage control?
  3. The frequency range for which the conductivity of Graphene in microwave frequency band is independent on frequency is (based on Fig. 1) much wider than 6 to 8 GHz. Why was such a narrow frequency range chosen?
  4. Which waveguide (WR?) was chosen for the measurement of the shielding effectiveness? What was its length and the mode of operation (TE ?, TM?)? Was it an SE analysis in the near or far field?
  5. Please provide a photo of the actual Multi-layer MIM structure, the shielding results of which are shown in Fig. 7.
  6. Line 180 "According to the previous paper" which article do the authors refer to?
  7. Fig.9 - Why two very low frequency values ​​were chosen for square and triangular waves? At higher values ​​one could observe an interesting effect of averaging the SE value.
  8. Figure 10 is too simple and obvious for a good research paper.
  9. The chapter "Conclusions" is treated very briefly and should be supplemented and expanded.

Author Response

  1. The size and resolution of Fig. 1 have been adjusted to make it clearer.
  2. According to ref [35], the carrier density of graphene ns can be modified by VDC between graphene and dielectric layer as ns = ε0εtVDC/te, where ε0 is permittivity of space, εt is the relative permittivity of the dielectric layer with thickness t, and e is the electron charge. Therefore, the increase of the thickness of the ion gel layer will theoretically lead to a higher voltage needed for the GIG structure to achieve the SE tuning. In this work, the ion gel was spin-coated onto the surface of graphene to acquire an uniform ion gel layer, in which way the size and thickness of ion gel layer was greatly limited. After squeezed and coated by the rolling machine, the GIG sample was obtained, and the thickness of ion gel layer was squeezed to about 50 μm, making it difficult to change its thickness and experimentally study how the shielding properties change as a function of the thickness of the Ion gel layer.
  3. The selection of the 6~8 GHz frequency band mainly considers the size of the sample and the limited experimental conditions. In this work, the obtained GIG sample is 8 x 8 cm with a total thickness of about 0.3 mm. For a sample of this size, it is difficult to directly test its S-parameters in a wide frequency range using the space-field method. Therefore, the 6-8 GHz frequency band is selected as a representative case, which is widely used in satellite communications and television broadcasting. In this band, a pair of waveguides can be used to directly measure the SE of the sample.
  4. The standard waveguide WR-137 was chosen standard waveguide WR-137, whose working frequency band was 5.38~17 GHz and its test window was 34.849 mm×15.799 mm. During the test, the operating mode of the electromagnetic wave in the waveguide was TE10, and the SE analysis was in the near field.
  5. The photo of the proposed Multi-layer MIM structure has been provided.
  6. I made an unclear expression here. Line 180 “the previous paper” refers to the principle analysis part of this article in section 2, rather than a specific article.
  7. When applying ion gel as the dielectric layer, since the moving speed of ions is far lower than that of electrons, it is difficult to achieve the rapid tuning speed of traditional electronic devices. Here two very low frequency values were chosen for square and triangular waves mainly to more intuitively observe the dynamic tuning capability of the GIG structure. When the frequency values further increase, the tunable range of SE with time gradually becomes smaller and tends to an average value.
  8. 10 shows the definition of time constant in the falling and rising process, and I will try to make it clearer.
  9. The chapter “Conclusion” has been expanded as suggested.

Reviewer 2 Report

The manuscript presents the study of shielding effectiveness in GHz and THz frequency bands using two types of graphene-based layered structures. The authors present arguments that their structures can be used as broadband voltage tunable shielding with switching times around one second. However, the language should be improved throughout the paper. It’s difficult to understand what the authors mean. The work should not be accepted in the current version before the below-presented issues are addressed.

The following major points should be addressed:

  1. The difference between Fig.5b and Fig.6a is not clear. Why the SE value at 0.5V in the first case (Fig.5b) is around 5dB, while in the second case (Fig.6a) it’s below 3dB? What is the reason for the change in SE voltage dependence character from nonmonotonic to linear ones? Also, the statement, that the MIM structure’s tuning range reaches higher values than the BIB structure’s is not obvious.
  2. How correct the THz SE spectra started from 0 THz? It seems like your THz SE spectra (Fig. 12) contains the GHz SE spectra. If so, there is a discrepancy between you SE values obtained by different methods, especially for the BIB structure.

Following minor points should be addressed:

  1. Absorbers (i.e., based on graphene) are well suitable for shielding applications, despite their resonant nature. That’s why I recommend referring in introduction articles devoted to the absorbers as a shielding [Zdrojek, Mariusz, et al. Nanoscale28 (2018): 13426-13431.], [Masyukov, M., et al. Journal of Optics22.9 (2020): 095105.]. 
  2. The abbreviation “PVDF-co-HFP and [EMIM][TFSI]” (line 111) should be expanded.
  3. Please specify the electron relaxation time of graphene used for conductivity calculation (Fig. 1).
  4. Specify the mass units for acetone solution (line 112).
  5. As a side remark, different colors in Fig.12 for the same voltage values are confusing.

Overall, in the paper, the complex study of shielding effectiveness in graphene-based structures has been performed, including operational speed and voltage tunability in different frequency bands. This fact is a key strength of the manuscript.

Author Response

  1. The reason for the difference between Fig.5b and Fig.6a is that the applying of bias voltage will change the p-doped energy level state of graphene exposed in air, making the Fermi level of graphene achieve a dynamic balance from the low doped state in the band gap center to the high doped state, and the change in SE voltage dependence character from nonmonotonic to monotonous correspondingly.
  2. In this paper, THz-TDS technology is adopted to test the SE of samples in THz band, whose frequency spectra is obtained by Fourier transform of measured ps-level time-domain terahertz pulse, which is not applicable for GHz band. Therefore, the measured SE in GHz band cannot accurately represent the shielding ability of samples against electromagnetic waves, and the results in GHz band should be removed.
  3. The recommended articles are of great help to me. In our work, the shielding of graphene for electromagnetic waves is mainly achieved by reflection. In my following research, I will consider the graphene-based absorber in shielding applications.
  4. The abbreviations have been expanded in the revised version.
  5. The electron relaxation time of graphene used for conductivity calculation is 0.2 ps.
  6. The mass units for the ingredients is gram.
  7. Thanks for reminding, the figure has been revised.

Round 2

Reviewer 1 Report

The paper was improved according to my comments and suggestions.

Reviewer 2 Report

The authors appropriately addressed the points raised in the comments.